# Distinctive Microbial Signatures and Gut-Brain Crosstalk in Pediatric Patients with Coeliac Disease and Type 1 Diabetes Mellitus

**DOI:** 10.3390/ijms22041511

**Published:** 2021-02-03

**Authors:** Parul Singh, Arun Rawat, Bara Al-Jarrah, Saras Saraswathi, Hoda Gad, Mamoun Elawad, Khalid Hussain, Mohammed A. Hendaus, Wesam Al-Masri, Rayaz A. Malik, Souhaila Al Khodor, Anthony K. Akobeng

**Affiliations:** 1Research Department, Sidra Medicine, Doha 26999, Qatar or pasi34086@hbku.edu.qa (P.S.); arawat@sidra.org (A.R.); baljarrah@sidra.org (B.A.-J.); 2College of Health & Life Sciences, Hamad Bin Khalifa University (HBKU), Qatar Foundation (QF), Doha 24404, Qatar; 3Division of Gastroenterology, Hepatology, and Nutrition, Sidra Medicine, Doha 26999, Qatar; ssundraraj@sidra.org (S.S.); melawad@sidra.org (M.E.); walmasri@sidra.org (W.A.-M.); aakobeng@sidra.org (A.K.A.); 4Department Medicine, Weill Cornell Medicine-Qatar, Doha 24144, Qatar; hyg2002@qatar-med.cornell.edu (H.G.); ram2045@qatar-med.cornell.edu (R.A.M.); 5Division of Endocrinology, Sidra Medicine, Doha 26999, Qatar; khussain@sidra.org; 6Division of General Pediatrics, Sidra Medicine, Doha 26999, Qatar; mrahal@sidra.org

**Keywords:** gut microbiota, T1DM, coeliac disease, children, pediatric neuropathy, corneal confocal microscopy

## Abstract

Coeliac disease (CD) and Type 1 diabetes mellitus (T1DM) are immune-mediated diseases. Emerging evidence suggests that dysbiosis in the gut microbiome plays a role in the pathogenesis of both diseases and may also be associated with the development of neuropathy. The primary goal in this cross-sectional pilot study was to identify whether there are distinct gut microbiota alterations in children with CD (*n* = 19), T1DM (*n* = 18) and both CD and T1DM (*n* = 9) compared to healthy controls (*n* = 12). Our second goal was to explore the relationship between neuropathy (corneal nerve fiber damage) and the gut microbiome composition. Microbiota composition was determined by 16S rRNA gene sequencing. Corneal confocal microscopy was used to determine nerve fiber damage. There was a significant difference in the overall microbial diversity between the four groups with healthy controls having a greater microbial diversity as compared to the patients. The abundance of pathogenic proteobacteria *Shigella* and *E. coli* were significantly higher in CD patients. Differential abundance analysis showed that several bacterial amplicon sequence variants (ASVs) distinguished CD from T1DM. The tissue transglutaminase antibody correlated significantly with a decrease in gut microbial diversity. Furthermore, the Bacteroidetes phylum, specifically the genus *Parabacteroides* was significantly correlated with corneal nerve fiber loss in the subjects with neuropathic damage belonging to the diseased groups. We conclude that disease-specific gut microbial features traceable down to the ASV level distinguish children with CD from T1DM and specific gut microbial signatures may be associated with small fiber neuropathy. Further research on the mechanisms linking altered microbial diversity with neuropathy are warranted.

## 1. Introduction

Type 1 diabetes (T1DM) [1] and coeliac disease (CD) [2] are two of the most frequent childhood autoimmune diseases [3,4]. T1DM is characterized by autoimmune destruction of β cells of the islets of Langerhans, causing insulin deficiency and hyperglycemia [5]. CD is triggered by exposure to gluten in genetically predisposed subjects resulting in small bowel mucosal atrophy and malabsorption [6]. The worldwide incidence of both T1DM [7], and CD [8,9] are increasing, and the prevalence of CD in children with T1DM is 5–7 times greater than that of the general population [3]. 

The coexistence of these two disorders has traditionally been attributed to the sharing of common high-risk human leukocyte antigen (HLA) (HLA) genotypes (DR-DQ) [3,10]. However, recent evidence suggests that environmental factors such as the composition of the gut microbiota plays an important role in the pathogenesis and co-occurrence of these disorders [11]. While intestinal dysbiosis has been clearly demonstrated in patients with CD [12,13,14,15], human data indicate subtle but significant differences in the gut microbiota of children with T1DM [16,17]. Despite the clinical overlap in their pathophysiology, little is known about the shared and distinct microbial characteristics in CD versus T1DM. The identification of novel microbial signatures (biomarkers) of T1DM and CD could be of enormous clinical importance.

Neuropathy occurs in patients with T1DM and CD and is more prevalent in those with both diseases [18,19,20]. Patients with T1DM and CD have a higher prevalence of retinopathy, nephropathy and neuropathy compared to patients with T1DM alone [18] and CD was found to be an independent risk factor for the development of retinopathy and nephropathy in a large cohort of patients with T1DM [21]. 

The pathogenesis of nerve damage in T1DM and CD is complex and is driven by both metabolic and inflammatory pathways. Recent studies have shown important bidirectional crosstalk linking the gut microbiota with the nervous system [22]. The focus of this association has been on the complex interaction between microbial metabolites, neuroendocrine system and immune system. Indeed, the emerging field of “psychobiotics” proposes to manipulate certain neurotransmitter-producing bacteria to regulate gut and neuronal function [23,24]. There are data linking an altered gut microbiome to endotoxin-mediated neurodegeneration [25]. More recently, gut microbiota has been linked to increased oxidative stress and inflammation leading to enteric neuropathy [26,27]. Several recent studies also suggest that altered gut microbiota composition may be involved in chemotherapy-induced peripheral neuropathy. [28,29]. In type 2 diabetes, a link between gut dysbiosis and metabolic as well as immune-mediated development of complications has been proposed [30] In a recent study, the abundance of Firmicutes and Actinobacteria was increased whilst Bacteroidetes was decreased in adults with type 2 diabetes and neuropathy [31]. Furthermore, fecal microbiota transplantation (FMT) has been used in autism, multiple sclerosis and Parkinson’s disease [32] and there is a single case of a 46-year-old female with type 2 diabetes reporting an improvement of painful diabetic neuropathy following FMT32 [33]. 

Corneal confocal microscopy (CCM) is a rapid, non-invasive and well-tolerated ophthalmic imaging technique that has been used to objectively quantify neurodegeneration in a wide range of peripheral and central neurodegenerative diseases [34,35,36]. We have also demonstrated significant corneal nerve loss in children with T1DM [37,38], with limited corneal nerve loss in children with CD [39].

In this study, we aimed to assess the gut microbiome composition in children with CD, T1DM and both T1DM and CD. We also assessed whether the composition was associated with the severity of corneal nerve loss as a biomarker of neurodegeneration. 

## 2. Results

### 2.1. Participants’ Characteristics

A total of 74 participants were enrolled in the study, of whom 13 were excluded for failure to provide the stool samples and another three participants were excluded based on the negative amplicon PCR results. A total of 58 participants were used for the final analysis (excluding the three positive controls): patients with CD (*n* = 19), patients with T1DM (*n* = 18), patients with both T1DM and CD (T1DM+CD, *n* = 9), and healthy controls (*n* = 12). Clinical and laboratory measures of patients in each group are summarized in (Table 1).

### 2.2. Richness and Diversity Analysis of the Gut Microbiota

A total of 5.08 million (5,079,867) paired-end sequences of the 16S rRNA gene were generated from 58 subjects. The mean number of the sequences was 58,784 ± 31,109 per sample. After denoising, 3016 Operational taxonomic units (OTUs) were obtained with the mean length 414.34 ± 17.57 bp. The OTUs were classified into a total of 14 different phyla as shown in the prevalence plot (Appendix A). The data was then normalized to overcome the inherent bias in amplicon sequencing, as discussed in the methods section. The entire microbial population were sufficiently represented as the rarefaction curves tapered phylogenetically with the increasing sequencing depth (Appendix A).

We found statistically significant differences in both the alpha (within the sample) and the beta diversity (between the samples overall microbial community structure) among the four groups. The overall structure of the fecal microbiota was evaluated using beta diversity and showed a clustering of samples according to the study group on an unweighted UniFrac dissimilarity matrix using PERMANOVA (Adonis test * *p* = 0.049) (Figure 1). Both CD and T1DM revealed compositional divergence from each other as well as from the healthy controls. The difference observed was statistically significant between patients with CD or T1DM+CD compared to healthy controls (** *p* = 0.006 and * *p* = 0.03 respectively). Interestingly, the T1DM+CD group overlapped with T1DM suggesting an overall compositional similarity between the two groups. These results indicate that CD and T1DM exhibit distinct microbial composition. In addition, the microbiome composition in CD patients accounted for a greater variability when compared to healthy controls.

Healthy controls showed a greater alpha diversity based on observed OTUs (**p* = 0.05) and marginally significant Chao1 indices (*p* = 0.066) (Figure 2). Patients with CD had markedly lower diversity, whereas those with T1DM had the highest diversity among the three disease groups (Figure 2). These results suggest that alterations in the gut bacterial microbiota diversity were aggravated in subjects with CD or with CD and T1DM.

Since the alpha diversity indices based on the observed OTUs were significantly lower in CD and T1DM+CD patients, we examined for a correlation between tTG-IgA levels (serological marker for CD) and the observed OTUs. Our data showed a significant negative correlation between the two parameters using regression analysis (* *p* = 0.03) (Figure 3). Additionally, we also performed correlation analysis of the observed OTUs with a subset of biochemical markers including Vitamin D, Folic acid and Vitamin B12 given their role in alteration of the gut microbial diversity [40,41,42,43]. However, none of them had a significant impact on diversity in our cohort (Appendix A).

### 2.3. Compositional Analysis of the Gut Microbiota

We assessed the compositional differences of the gut microbiota at the phylum and family levels. In all the study groups, the bacterial gut microbiota was dominated by bacteria that belong to Firmicutes, Bacteroidetes, Actinobacteria, Proteobacteria phyla (Figure 4A) and the *Lachnospiraceae*, *Ruminococcaceae*, *Bacteroidaceae* and *Prevotellaceae* families (Figure 4B). No statistical differences were observed between the four groups at the phylum or family levels (Appendix A).

Given the known association of gram negative bacteria in the pathogenesis of CD [44], we compared the levels of the pathogenic proteobacteria in all groups. Consistent with previous studies, our data showed that the levels of *Shigella/E.coli* were significantly higher in the CD group compared to healthy controls (Figure 5).

We then used DeSEQ2 to determine if specific bacterial signatures were associated with the disease phenotype. A total 42 differentially abundant bacterial ASVs were identified when comparing the disease groups to healthy controls (Appendix A).

Compared to the other groups, CD patients exhibited an exclusive microbial signature characterized by a decrease in *Dialister* and *Parabacteroides* and an increase in *Ruminococcus* (Table 2). Similarly, a significant increase in *Akkermansia* and *Barnesiella* was found exclusive in T1DM. Interestingly, a decrease in *Coprococcus* genus levels was a hallmark in the gut microbiome of patients with both T1DM and CD (Table 2).

On the other hand, 7 ASVs were found to be common between the disease groups compared to healthy controls (Table 3). There were striking differences in the pattern of abundance of the ASVs, with ASVs in the CD group exhibiting an opposing trend compared to T1DM and CD+T1DM groups (Table 3).

Collectively, this data indicated that children with CD carry a distinctive gut microbial signature characterized by an increase of *Ruminococcus*, *Prevotella* (Prevotella_1, Prevotella_3, Prevotella_4), *Alistipes*, *Oscillospira* (Oscillospira_2), *Clostridium* (Clostridium_2), and a decrease in *Parabacteroides*, *Dialister* and *Bacteroides* (Bacteroides_1). On the other hand, patients with T1DM showed an increase in *Bacteroides* (Bacteroides_1), *Akkermansia, Barnesiella* and a decrease in *Prevotella* (Prevotella_1, Prevotella_3, Prevotella_4), *Alistipes*, *Oscillospira* (Oscillospira_2), *Clostridium* (Clostridium_2). Patients with T1DM and CD showed a signature common to T1DM patients except for an exclusive decrease in *Coprococcus*.

### 2.4. Association of the Microbial Signatures with Corneal Nerve Fiber Damage

Knowing that neuropathy is a known complication associated with T1DM and CD, and that a possible bidirectional crosstalk linking the gut microbiota with the nervous system has been previously described [22], we wanted to test whether the gut microbes identified in this study were correlated with corneal neuropathy. We have shown that CCM demonstrates evidence of neuropathy in these children with T1DM [38] and to a lesser extent in those with CD [39]. We performed a linear regression to determine any possible association between the abundance of the gut microbiota and abnormal corneal nerve fiber length (CNFL) (Appendix A). Our data showed that Bacteroidetes phylum (Figure 6A) specifically *Parabacteroides* genus (Figure 6B) was negatively correlated with CNFL, (*p* = 0.03625 and 0.00179 respectively). Vitamins D and B have been previously reported to play a potential role in the pathophysiology of neuropathies [45,46,47,48], however in our cohort we did not observed a significant correlation between these biochemical markers and neuropathy (Appendix A).

### 2.5. Predicted Gut Microbiota Functions

To understand the possible link between the microbial composition and neuropathy, we conducted Phylogenetic Investigation of Communities by Reconstruction of Unobserved States (PICRUSt v1.0.0) analysis to determine the predicted metabolic functions of the microbial communities. Our data showed that specific pathways such as Retinol metabolism, Biosynthesis of unsaturated fatty acid (Figure 7), ethylbenzene degradation (Appendix A) were significantly downregulated in patients with neuropathy.

## 3. Discussion

An imbalance of the gut microbiota composition, called microbial dysbiosis, is characterized by decreased microbial diversity, gain or loss of specific community members or changes in their relative abundance [49]. A wide array of studies have identified significant differences in the microbial diversity and taxonomic composition of gut microbial communities between healthy controls and patients with CD [44,50,51,52] or T1DM [53,54,55]. However, the specific microbial taxonomic differences vary widely depending on the study design, and a definitive disease-associated community structure has not been identified. This may be due to the large variation in the gut microbial community composition among subjects and/or to technical differences among the studies.

To our knowledge, this is the first study to have directly compared the gut microbial signatures of children with CD, T1DM, T1DM+CD and healthy controls, with the aim to uncover the shared and distinct microbial ASVs associated with these diseases.

In our study, CD subjects showed an increased abundance of *Ruminococcus* and decreased abundance of *Parabacteroides* and *Dialister.* Enriched *Ruminococcus* is associated with irritable bowel syndrome (IBS) [56], transient blooms of pro-inflammatory *Ruminococcus* have been associated with active flare-ups in IBD [57] and has been found to secrete a unique L-rhamnose oligosaccharide that induces tumor necrosis factor alpha (TNF-a), a major pro-inflammatory cytokine [58]. Studies have also reported a decreased abundance of *Parabacteroides* genera in IBD and Crohn’s disease patients, respectively [59,60]. Genus *Dialister* is capable of generating both acetate and propionate [61], and its abundance is reduced in patients with Crohn’s disease [62]. These results suggest that the exclusive microbial signatures of CD overlapped with those of other functional gastrointestinal disorders.

The exclusive signatures associated with T1DM were reflected as an increase in the abundance of *Akkermansia* and *Barnesiella*, both of which are gram-negative lipopolysaccharide (LPS)-producing bacteria [63]. The LPS released by these bacteria may mediate inflammation, obesity and insulin resistance [64]. In the past, populations of *Barnesiella spp*. were found to be increased in obese prediabetic mice as well as in the presence of diet-induced diabetogenic intestinal environment [65].

As mentioned in the results there were seven ASVs that were common between the three diseased groups (CD/T1DM/T1DM+CD). Interestingly these ASVs showed a completely opposing trend between the CD and T1DM patients suggesting that though both the diseases have shared genetic background and are characterized by a decrease in microbial diversity, distinct alterations in specific bacteria could drive disease causation. Specifically, we observed the abundance of *Bacteroides* (ASV2) decreased in CD while increased in both T1DM, T1DM+CD. Conversely, *Prevotella* (ASV 4,14,21), *Alistipes* (ASV 5), *Oscillospira* (ASV 8), *Clostridium* (ASV 11) all increased in CD, while decreased in T1DM and T1DM+CD groups.

The concomitant presence of T1DM with other autoimmune disorders is referred to as autoimmune polyendocrine syndromes (APS) [66]. The coexistence of T1DM and CD is a subtype of APS-4 [67,68]. To our knowledge, no study thus far has examined the gut microbial signatures in patients with concomitant T1DM+CD (APS-4 subtype). In our study we found that among the common ASVs between the three diseased groups, subjects with concomitant T1DM+CD showed the same trend as T1DM. This is interesting to note as studies have reported that T1DM diagnosis precedes CD diagnosis in about 90% of patients [69]. The recent Teddy (The environmental determinants of diabetes in the young) study that analyzed 5891 children at high genetic risk of both diseases found that T1DM autoimmunity usually precedes CD autoimmunity and the T1DM-associated islet autoantibodies (IAs) significantly increase the risk of subsequent CD-associated tissue transglutaminase autoantibodies [70]. Prospective studies, such as the Type 1 Diabetes Prediction and Prevention study in Finland [71], BABYDIET in Germany [72], and DIABIMMUNE study in Finland, Estonia, and Russia [16,73] have shown that gut microbial dysbiosis starts prior to the appearance of islet autoantibodies and T1DM onset. A general trend identified is the increased abundance of the genus *Bacteroides* and clear decreases in diversity [54]. Both these trends are in line with our findings and could explain the deviation of the intestinal microbiota in a similar way in both T1DM and T1DM+CD groups as the dysbiosis could be driven by the autoimmune microbiota of T1DM which is different from that of children with CD and healthy controls.

Recent studies have shown that histological duodenal damage in patients with coeliac disease correlates with presence of high tTg-IgA titers [74,75,76,77]. Many of these studies thus propose that in cases with strong positive tTG levels, duodenal biopsies could be omitted. In our study for the first time we found a significant negative correlation between the tTG titer and gut microbial diversity. Generally, a healthy, immune-resilient and stable gut relies on high microbiota richness and biodiversity [78]. Thus, we suggest that gut microbiota diversity could also be used as an additional criterion for improving the predictive value of tTG levels for CD diagnosis.

The gut-brain axis is an information exchange platform which allows two-way communication between the gut and the host nervous system [22]. Information can be exchanged via neural network, hormones and the immune system [79]. Disruption of the delicate balance between the host and gut bacteria could be a contributing factor behind many neurological diseases [80]. Despite the vast literature supporting the link between the gut microbiota and the nervous system, few studies have examined the impact of perturbation in specific gut bacteria on neuropathy.

Corneal nerve loss has good diagnostic utility for both diabetic somatic and autonomic neuropathy [35]. In our cohort we found a significant negative correlation between neuropathy (as determined by CNFL) and *Parabacteroides* (Phylum Bacteroidetes) abundance. Previous studies have reported reduction of the *Parabacteroides* in multiple sclerosis patients where its exerts a protective role by stimulating anti-inflammatory IL-10–expressing human CD4^+^CD25^+^ T cells [81]. In a study assessing the neuroprotective ability of fifty gut bacterial strains, *Parabacteroides distasonis* had the strongest capacity to reduce IL-6, with a potential to be used as a live biotherapeutic in disorders characterized by neurodegeneration and neuroinflammation [82]. Oral administration of *Parabacteroides* components (mPd) is known to be effective in dampening the production of proinflammatory cytokines via innate and adaptive immunomodulatory mechanisms [83]. Our results suggest *Parabacteroides* may play a protective role against neuropathy, and its decreased abundance may be associated with greater corneal nerve loss and hence neuropathy.

We analyzed the potential function of gut microbiota based on PICRUSt analysis.

The neuropathic groups showed significantly decreased microbial function related to biosynthesis of unsaturated fatty acids, retinol metabolism and ethylbenzene degradation in comparison to controls. Neurons cannot synthesize long-chain fatty acids but can incorporate them in their membranes [84]. The regulation of synthesis and release of pro-inflammatory mediators such as interleukin-1 beta (IL-1β), interleukin-6 (IL-6) and tumor necrosis factor alpha (TNF-α) by unsaturated fatty acids is known to play a role in neuronal plasticity [85,86]. Similarly, retinoid acid modulates neural patterning, differentiation, axon outgrowth, neuronal functioning, signaling and neuronal plasticity [87,88]. Impaired retinoic acid signaling results in neuroinflammation, oxidative stress, mitochondrial dysfunction, and neurodegeneration [89]. Thus, the reduction in unsaturated fatty acids and retinol biosynthesis may exacerbate inflammatory responses and lead to neuronal damage [90,91]. The significance of ethylbenzene degradation in neuropathy is mostly unclear and remains to be determined.

A major limitation of the current study is the small cohort size of the different disease groups. Nevertheless, the detailed microbiota composition and functional analysis in relation to phenotyping paves the way for the development of a set of microbiota-based biomarkers to stratify children with CD and T1DM. It also highlights the association between the gut microbial diversity and tTG-IgA as a promising alternative for the evaluation of CD. Finally, this study suggests a potential role for specific gut bacteria in the development of sub-clinical neuropathy in pediatric patients with CD and T1DM.

## 4. Methods

### 4.1. Study Participants and Design

The study was approved by Sidra Medicine IRB (protocol# 1708012783). The investigators ensured that the study was conducted in full conformity with the current version of the Declaration of Helsinki and with the ICH Guidelines for Good Clinical Practice (CPMP/ICH/135/95) July 1996. 

Children with a diagnosis of T1DM and/or CD attending the Sidra Medicine’s outpatient clinic were invited to participate in the study. In addition, age-matched children (without T1DM or CD) served as healthy controls. All participants provided informed consents/assents before inclusion. Inclusion criteria were patients aged between 8 and 18 years; diagnosis of CD or T1DM by established criteria; parental and child’s consent to participate in the study. Exclusion criteria were patients with any known disease of the cornea or history of trauma or surgery to the cornea; patients with any known cause of peripheral neuropathy other than T1DM or CD; patients with any serious chronic illness that might affect the cornea and/or nervous system; patients taking antibiotics (for the last 6 months), failure to achieve informed consent from parent/guardian and/or assent from the adolescent. 

### 4.2. Data Collection

Basic data was collected from all patients including age, gender, nationality, diagnosis (T1DM, CD, T1DM+CD), time since diagnosis (months) and co-morbidities. All participants had their height and weight recorded. Height was recorded to the nearest mm using a standard office stadiometer. Weight was recorded to the nearest 0.1 kg using a standard office scale. Body mass index (BMI), weight and height z scores were calculated based on gender and age-related centile charts.

All participants with T1DM and/or CD had blood samples collected to assess complete blood count (CBC), liver function tests (LFTs), thyroid function tests (TFTs), folic acid, iron profile, ferritin, and vitamin B12. In addition, patients with T1DM/T1DM+CD had the HbA1C checked and those with CD/T1DM+CD had the tissue transglutaminase IgA antibodies(tTg) measured. No blood samples were taken from control participants specifically for this study.

### 4.3. Corneal Confocal Microscopic Examination 

Using the Heidelberg Retina Tomograph Cornea Module (Heidelberg Engineering, Vista, CA, USA), participants underwent corneal confocal microscopy as per an established protocol [38]. Briefly, one eye was randomly selected and anaesthetized with a drop of Alcaine (proparacaine hydrochloride 0.5%; Alcon, Mississauga, ON, Canada). A drop of Genteal Gel (0.3% hypromellose; Novartis Ophthalmics, Mississauga, Ontario, Canada) was placed on the tip of the objective lens and a sterile disposable Tomo cap was placed over the lens, allowing optical coupling of the objective lens to the cornea to focus on Bowman’s layer. Images were taken by a single investigator. Five images per participant were examined and nerve fiber density (number of nerve fibers per mm [2] of corneal tissue), nerve branch density (number of branches from major nerve trunks per mm [2] of corneal tissue), nerve fiber length (length of all nerve fibers per mm [2] of corneal tissue) and nerve fiber tortuosity were quantified. 

### 4.4. Microbial DNA Extraction 

A fraction of the stool sample collected from the study participants (400–500 mg) was transferred to the OMNIgene GUT kit (DNA Genotek Inc, Ottawa, ON, Canada), according to manufacturer’s protocol. Fecal DNA extraction was performed using the QIAamp Fast DNA Stool Mini Kit as per manufacturers’ instructions. The DNA concentration and purity were evaluated using a Nanodrop^®^ spectrophotometer (Thermo Scientific, Wilmington, DE, USA). The extracted DNA samples were stored at −20 °C until library preparation.

### 4.5. 16 S ribosomal RNA Gene Amplification and Illumina Sequencing

The bacterial 16S rRNA variable 16 rRNA gene fragments V3 and V4 were amplified with polymerase chain reaction (PCR), using the Illumina recommended amplicon primers: 

Forward: 5′TCGTCGGCAGCGTCAGATGTGTATAAGAGACAGCCTACGGGNGGCWGCAG; 

Reverse: 5′GTCTCGTGGGCTCGGAGATGTGTATAAGAGACAGGACTACHVGGGTATCTAATCC. The amplifications were performed under the following conditions: initial denaturation at 95 °C for 2 min, followed by 30 cycles of denaturation at 95 °C for 30 s, primer annealing at 60 °C for 30 s, and extension at 72 °C for 30 s, with a final elongation at 72 °C for 5 min. Three positive controls were also included along with the study samples. Amplicons were purified according to the Illumina MiSeq 16S Metagenomic Sequencing Library Preparation protocol (http://support.illumina.com/downloads/16s_metagenomic_sequencing_library_preparation.html). A second PCR step was used to multiplex purified amplicons using a dual-index approach with the Nextera XT index kit (Illumina, San Diego, CA, USA). Cycle conditions were 95 °C for 3 min, 8 cycles; 95 °C for 30 s, 55 °C for 30 s, 72 °C for 30 s, and then 72 °C for 5 min. Amplicon generation was validated through visualization on a 2% (*w*/*v*) agarose gel. Amplicon library concentrations were determined using the Qubit HS dsDNA assay kit (Life Technologies, Australia). The final library was paired end sequenced at 2 × 300 bp using a MiSeq Reagent Kit v3 on Illumina MiSeq platform (Illumina, San Diego, CA, USA).

### 4.6. Microbiome Sequence Data Processing and Diversity Analysis

Read 1 and Read 2 of 16S rRNA gene from all amplicons were quality checked with FastQC (http://www.bioinformatics.babraham.ac.uk/projects/fastqc). The paired end demultiplexed sequences were imported into Quantitative Insights into Microbial Ecology (QIIME2; version 2019.4.0) software package [92,93] (https://qiime2.org/). DADA2 [94] was used to denoise the data and execute different steps like read filtering, de-replication and chimera removal. The paired-end reads were trimmed both from forward and reverse end, and a minimum read length of 250bp was used for further processing to generate the amplicon sequence variant (ASV). Taxonomic classification was performed using the 16S rRNA gene database from Greengenes (http://greengenes.lbl.gov) (version 13_8) [95]. The taxonomic classification of ASVs was done by using pre-trained classifier (a scikit-learn naive Bayes machine-learning classifier) against Greengenes database 13_8 (99% OTU full-length sequences) as provided by Qiime2 (https://docs.qiime2.org/2020.2/data-resources/) [96]. The data was then imported into R (RStudio v 1.2 with R v 3.6) as Biological Observation Matrix (biom). The data was further evaluated with the Phyloseq package [97]. The final set of ASVs were used for taxonomical classification.

Alpha Diversity (within sample community) was assessed by different indices like Shannon [98], Chao1 [99], observed OTUs using the R package “vegan” (v2.5–6) [100]. Beta Diversity (Divergence in community composition between samples) was assessed by using four different distance metrics Bray-Curtis (abundance) and Jaccard, Weighted Unifrac, Unweigted Unifrac.

### 4.7. Canonical Correspondence Analysis

Canonical Correspondence Analysis (CCA) was used as an ordination method, and significance was determined using Adonis test (PERMANOVA) that considers the multidimensional structure of the data (e.g., compares entire microbial communities) (999 permutations).

### 4.8. DESeq2 Differential Abundance Analysis

We used DESeq2 [101] to delineate the differentially expressed bacterial taxa responsible for different disease states. DESeq2 has been shown to perform well when applied to uneven library sizes and sparsity common in 16S rRNA gene marker data. The analysis was carried out on the un-rarefied data as the in-build library size normalization allows maximum participation of sequenced reads (taking the entire data into consideration).

### 4.9. Functional Analysis

Metagenome functional contents were analyzed using the PICRUSt software package (v1.0.0) to predict gene contents and metagenomic functional information [102]. The statistical evaluation was then performed with STAMP [103] and significant pathways (*p*-value < 0.05, CI 99%) were exported.

### 4.10. Statistical Analysis

We used Kruskal-Wallis and Wilcoxon signed rank test to perform nonparametric statistical test [104], we calculated the false discovery rate (FDR) with Bonferroni correction and the resulting *p*-value < 0.05 was considered significant for all tests. Additionally, a simple linear regression and linear mixed models were used for correlation analysis.

## Figures and Tables

**Figure 1 ijms-22-01511-f001:**
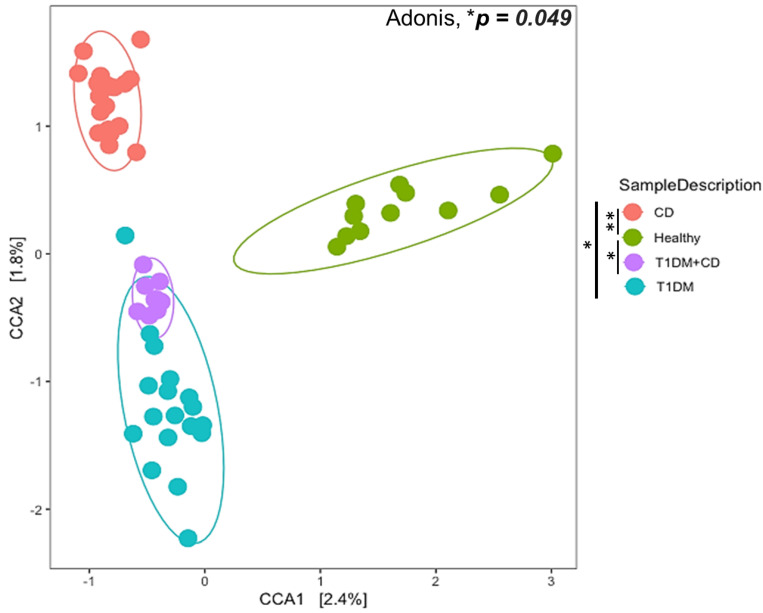
Ordination plot of Canonical Correlation Analysis (CCA) to explore between the sample diversity (Beta diversity). Samples from different groups: CD (orange), T1DM (blue), healthy control (green), T1DM+CD (purple) had a significant overall difference in composition (Adonis * *p* = 0.049). The difference observed was statistically significant between patients with CD or T1DM+CD compared to healthy controls (** *p* = 0.006 and ** p* = 0.03 respectively). Abbreviations: T1DM = Type 1 Diabetes; CD = Coeliac disease; CCA = Canonical Correlation Analysis.

**Figure 2 ijms-22-01511-f002:**
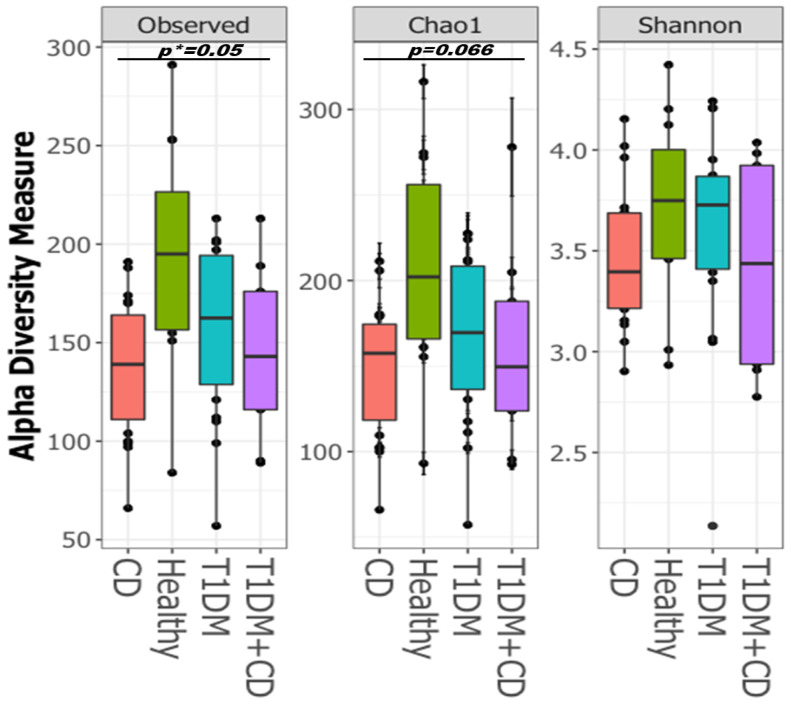
Boxplots of Alpha-diversity indices: Observed OTUs; Chao1; Shannon. Boxes represent the interquartile range (IQR) between the first and third quartiles (25th and 75th percentiles, respectively), and the horizontal line inside the box defines the median. Whiskers represent the lowest and highest values within 1.5 times the IQR from the first and third quartiles, respectively. Statistical significance between the sample groups CD (orange), T1DM (blue), healthy control (green), T1DM+CD (purple) was identified by the Kruskal-Wallis test with false discovery rate (FDR)-Bonferroni corrected *p* values. * *p* ≤ 0.05). Abbreviations: T1DM = Type 1 Diabetes; CD = Coeliac disease.

**Figure 3 ijms-22-01511-f003:**
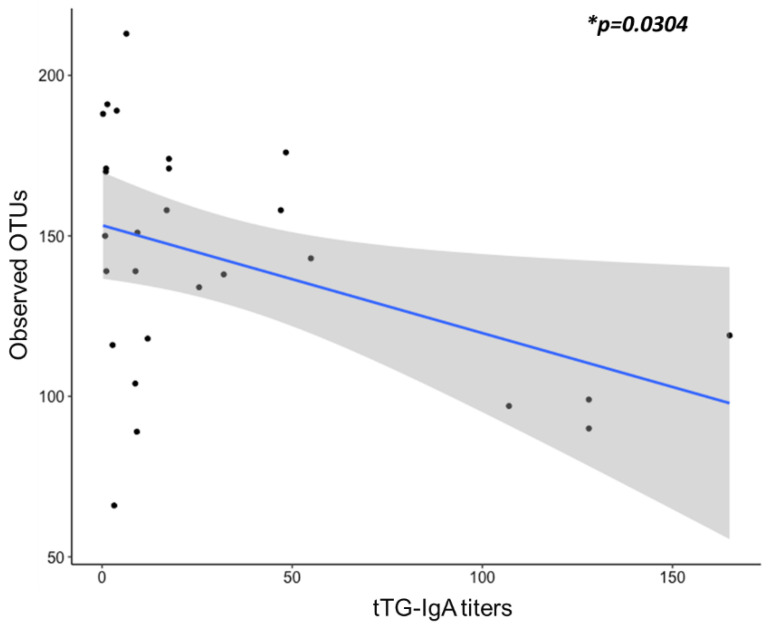
Linear regression was used to determine the correlation between the tTG titers (*x*-axis) and alpha diversity indices observed OTUs (*y*-axis). Abbreviations: OUT = Operational taxonomic units; Ttg-IgA = tissue transglutaminase IgA antibodies.

**Figure 4 ijms-22-01511-f004:**
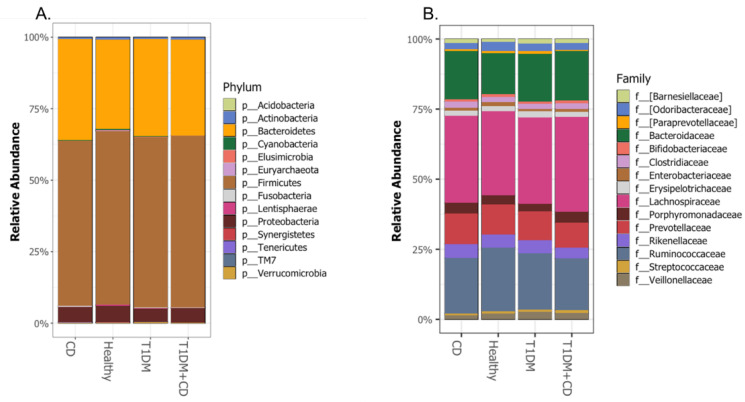
Comparison of the relative abundance at the (**A.**) Phylum (**B.**) Family levels among the different groups included. Abbreviations: T1DM = Type 1 Diabetes; CD = Coeliac disease; p = phylum; f = family.

**Figure 5 ijms-22-01511-f005:**
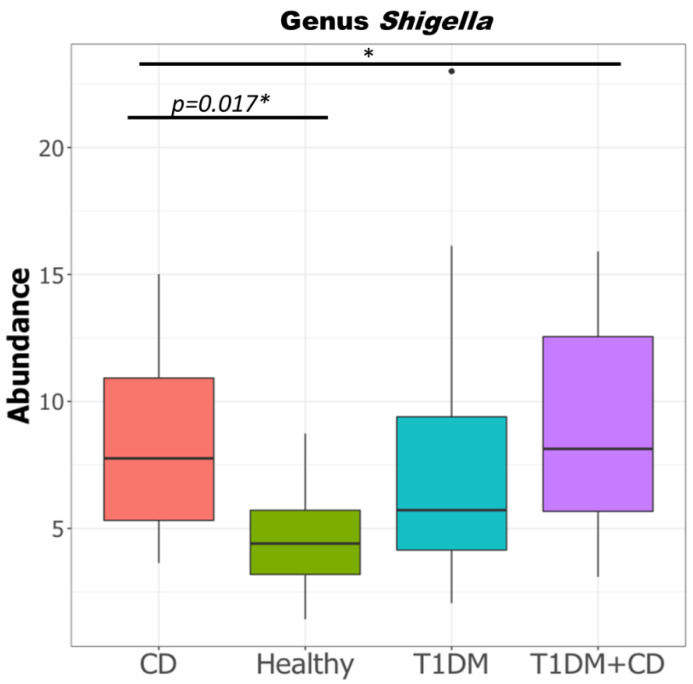
Difference in relative abundance of genus *Shigella* in the four study groups CD, T1DM, healthy control, T1DM+CD. (Kruskal-Wallis test with false discovery rate (FDR)-Bonferroni corrected *p* values. * *p* ≤ 0.05). Significant difference was observed across the four groups as well as between CD and healthy controls. Abbreviations: T1DM = Type 1 Diabetes; CD = Coeliac disease.

**Figure 6 ijms-22-01511-f006:**
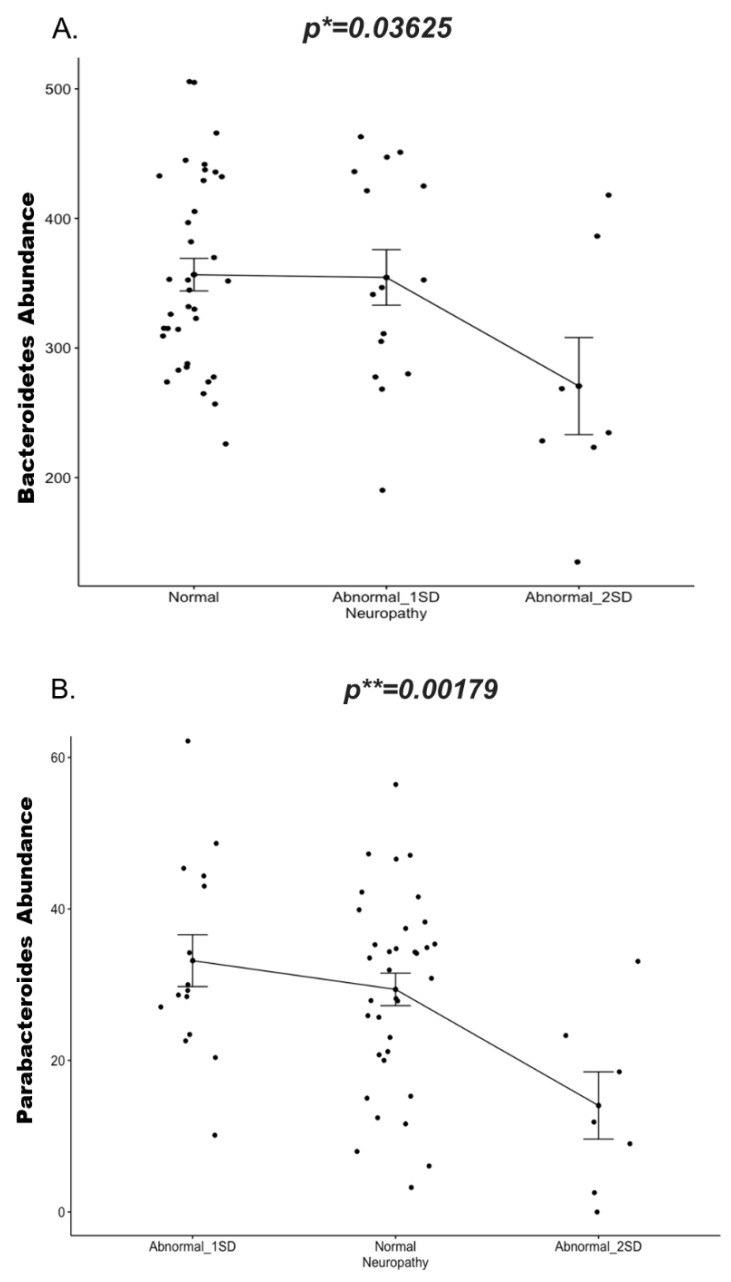
Change in abundance (*y*-axis) of (**A.**) Phylum *Bacteroidetes and* (**B.**) Genus *Parabacteroidetes* in the neuropathy group(*x*-axis) at two standard deviations (1SD and 2SD lower than the mean of controls) compared to normal subjects (without neuropathy). Significance is determined by the linear mixed model. Corrected *p* values are indicated on the graph. * *p* ≤ 0.05; ** *p* ≤ 0.01). Abbreviations: SD = Standard deviation.

**Figure 7 ijms-22-01511-f007:**
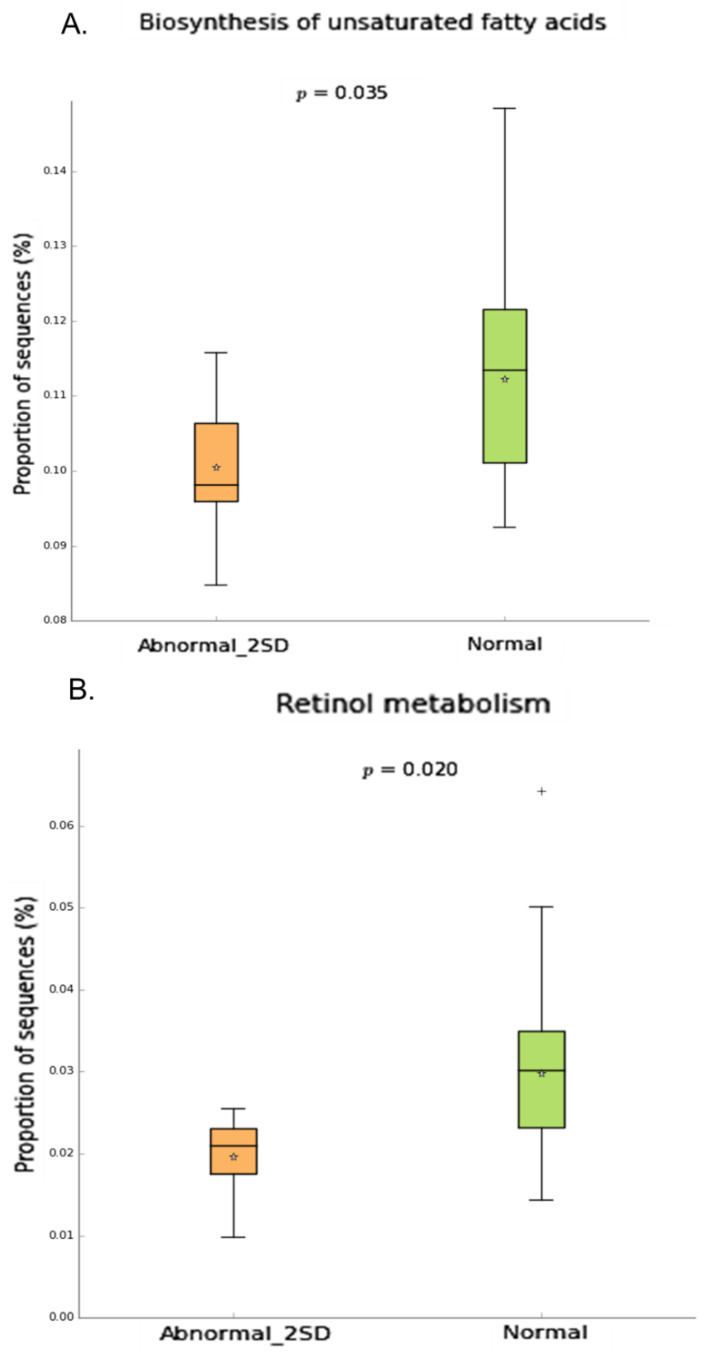
Inferred gut microbiome functions by PICRUSt from 16S rRNA gene sequences in neuropathy(2SD) compared to normal group. Difference in predicted functions of genes involved in (**A.**) biosynthesis of unsaturated fatty acids (**B.**) Retinol metabolism. (Mann-Whitney* *p* < 0.05;). Abbreviations: SD = Standard deviation.

**Table 1 ijms-22-01511-t001:** Basic and clinical demographics in control subjects, CD, T1DM, T1DM+CD patients.

	Healthy Controls (*n* = 12)	T1DM (*n* = 18)	CD (*n* = 19)	T1DM+CD (*n* = 9)
Age (years)	13.6 ± 1.3	15.1 ± 2.5	12.7 ± 1.8	14.1 ± 2.6
Disease duration(years)	-	5.1 ± 4.9	-	6.4 ± 3.7
BMI (kg/m [2])	20.26 ± 4.2	21.5 ± 5.3	18.8 ± 3.8	21.8 ± 4.6
Tissue Transglutaminase IgA antibodies(tTg) (unit/mL)	-	-	26.7 ± 52.8	30.9 ± 42.3
Hemoglobin (g/L)	127.30 ± 9.6	140.6 ± 11.1	129.2 ± 8.4	130.2 ± 19.3
Platelets(×10 [9]/L)	348 ± 91.1	290 ± 49	321.7 ± 80.2	331.2 ± 87.8
25 OHD (nmol/L)	53.6 ± 13.9	45.7 ± 22.8	45.7 ± 13.8	50.8 ± 9.3
Vitamin B_12_ (ng/mL)	-	-	376.6 ± 110	237.3 ± 136.7
Folic acid (nmol/L)	-	-	44.05 ± 1.45	41.5 ± 62.3
Serum Iron (μmol/L)	-	9.5 ± 2.6	11.0 ± 0	8.8 ± 4.7

Data are presented as mean ± SD, vitamin B_12_ and folic acid levels were compared with the normal laboratory range. Notes: Reference ranges: tissue transglutaminase, <7; vitamin D, 75–200; vitamin B12, 149–772; folic acid, <6.8; serum iron, 3.1–23.1. Abbreviations: T1DM = Type 1 Diabetes; CD = Coeliac disease; BMI = Body mass index; 25 OHD = 25-hydroxycholecalciferol.

**Table 2 ijms-22-01511-t002:** Differentially abundant ASVs exclusive to each of the diseased groups (CD, T1DM, T1DM+CD) compared to healthy controls. **↑** (increased abundance relative to healthy control) **↓** (decreased abundance relative to healthy control) Abbreviations: T1DM = Type 1 Diabetes; CD = Coeliac disease; ASV = Amplicon sequence variant.

ASV ID	ASVs	Phylum	Genus	CD	T1DM	T1DM+CD
**ASV 3**	e070c6e413d129c7da1d7eed09996432	Bacteroidetes	*Parabacteroides*	↓		
**ASV 6**	d49b8bc15a16fdcec90109b5ffba4545	Firmicutes	*Ruminococcus*	↑		
**ASV 17**	4bc96a49376733f3cef3324a92407e80	Firmicutes	*Dialister*	↓		
**ASV 29**	53f18255a6dbd7919061166c2106bf9a	Verrucomicrobia	*Akkermansia*		↑	
**ASV 32**	147f8f2bc7eb19ffd9ca67b7248b88e6	Bacteroidetes	*Barnesiella*		↑	
**ASV 41**	071d6c95a1ffe99d059133f97a6babbe	Firmicutes	*Coprococcus*			↓

**Table 3 ijms-22-01511-t003:** Differentially abundant ASVs that were common among the three diseased groups (CD, T1DM, T1DM+CD) compared to healthy controls. **↑** (increased abundance relative to healthy control) **↓** (decreased abundance relative to healthy control). Abbreviations: T1DM = Type 1 Diabetes; CD = Coeliac disease; ASV = Amplicon sequence variant.

ASV ID	ASVs	Phylum	Genus	CD	T1DM	CD+T1DM
**ASV 2**	e88e8770d92f876d5761d44eb18b48b0	Bacteroidetes	*Bacteroides_1*	↓	↑	↑
**ASV 4**	dce9684604d7dca4d75ec4cd6fedca3e	Bacteroidetes	*Prevotella_1*	↑	↓	↓
**ASV 5**	d9789dd557d546b1d6c064b18c6258e9	Bacteroidetes	*Alistipes*	↑	↓	↓
**ASV 8**	bbb86211c2d4d6af9e76ef5d03919cfc	Firmicutes	*Oscillospira_2*	↑	↓	↓
**ASV 11**	a2ec826da8c928b32e47fd1faa9c41d9	Firmicutes	*Clostridium_2*	↑	↓	↓
**ASV 14**	76ff6e86449f4f664ac157d47c6910f5	Bacteroidetes	*Prevotella_3*	↑	↓	↓
**ASV 21**	0627f945c2442b75d2b1ca46c6bb0cc5	Bacteroidetes	*Prevotella_4*	↑	↓	↓

## Data Availability

The data is available upon request.

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
