# Peer review of "Distinctive Microbial Signatures and Gut-Brain Crosstalk in Pediatric Patients with Coeliac Disease and Type 1 Diabetes Mellitus"

_ijms, 2021, doi:10.3390/ijms22041511_

Round 1

Reviewer 1 Report

The manuscript by Singh et al. is an interesting cross-sectional investigating the gut microbiota composition and biodiversity in pediatric patients with type 1 diabetes and/or celiac disease, comparing them with age-matched healthy controls. Authors also examined the relationship between   gut microbiome composition and neuropathy, as assessed by the presence of corneal nerve fiber damage at the corneal confocal microscopy. Interestingly, authors found disease-specific gut microbial features that may be useful to distinguish children with CD from T1DM. In addition, authors suggest that specific gut microbial signatures may be associated  with small fiber neuropathy in pediatric subjects with T1D and CD.

Additional analyses may add significant values to this interesting paper. Some clarifications are required. In addition, moderate English improvement is needed (mainly regarding the structure of many sentences). My comments are as follows:

-Title should include a reference to “neuropathy assessed by corneal confocal microscopy”; the same applies to keywords

-Keywords: amend “pediatrics”

-Authors define this study as “prospective cross-sectional study” but it seems that patients were evaluated only at a given time point; therefore, the study definition should be amended into “cross-sectional study”. Also, authors need to be more informative on the disease duration in each disease group. Are all the study participants newly diagnosed with T1D and/or CD? Does the study include a mixed cohort of participants with both new-onset and long-standing T1D/CD? Please, clarify.

Line 39: remove the extra space

Line 41: specify in which group(s)

Line 54: ref 3 is quite outdated; authors should add this PMID: 30045024

Line 57: insert a comma between the two references 7 and 8

Line 59: “coexistence” rather than “concomitance”

Line 60: amend “human leukocyte antigen (HLA)”

Line 68: remove the comma after “neuropathy”

Line 69: add this reference PMID: 30759885

Line 76: amend “between microbial metabolites, neuroendocrine system and immune system”

Line 79: “to endotoxin-mediated neurodegeneration”

-Line 80: altered gut microbiome composition?  // Line 80: gut microbiota composition // Line 81: altered gut microbiota composition

Lines 80-81: “inflammation leading to enteric neuropathy”

Line 82: “chemotherapy-induced”

Lines 82-82: “In type 2 diabetes, a link between gut dysbiosis and metabolic as well as immune-mediated development of complications has been proposed”

Line 84: “abundance” instead of “richness”;; also “, whilst Bacteroidetes abundance…”

Line 87: “there is a single case of a 46-year-old female with type 2 diabetes reporting an improvement of painful diabetic neuropathy following FMT32”

Line 101: outpatient Clinic

Line 103: “provided informed consent/assent” instead of “were provided…”

Line 113: gender

Line 116: Body mass index (BMI)

Line 133: mm2

Line 218: A total of 74 participants

-Table 1: control column: modify “Healthy controls”;; specify (here and in the methods) “Tissue Transglutaminase IgA antibodies”; in the table, specify the unit of measurement of tTg-IgA;; there should be something wrong in the hemoglobin values, please check; indicate platelet unit of measurement (x103/mL);; indicate reference ranges for each parameter established by your Institution lab in another column or in Table footnotes.

-Table 1: authors should indicate disease duration for each disease group

-Authors should provide information on statistical comparison and statistical differences between groups with regard to parameters listed in Table 1; In addition, regression analysis should be performed even to investigate the correlation between observed OTUs (and nerve loss) and different parameters listed in Table 1, particularly vitamin D, vitamin B12 and folic acid (given the potential role of such vitamins in the pathophysiology of neuropathies: see and cite PMID: 30531087; PMID: 25299283 ; PMID: 32767343 ; PMID: 32825324)

Line 234: amend the term “biasness”

-Figure 1: indicate the p-value for each asterisk symbol * and ** and the group comparison

-Figure 2: I can’t find the double **(p<=0.01) in the Figure

-Figure 3: add “Observed OTUs” in the y-axis and “tTG-IgA titers” in the x-axis

-Figure 5: specify what do the upper black line and its asterisk stand for

-For each figure and table, authros should always write in full all the abbreviations in footnotes or legends.

Line 329: disease groups

-Line 350: correlated with corneal neuropathy

-Please, list the figures and tables in a sequential manner with respect to the order in which they appear in the text; also, I cannot find suppl. Figure 8 in the supplementary appendix

-Lines 371-376: write PUFA in full first and then abbreviate; remove brackets between “retinol” and “ethylbenzene degradation”; in Figure 7, a section showing ethylbenzene degradation is missing.

-Figure 7 legend (compard to main text): do author mean PUFA (polyunsaturated) or unsaturated fatty acid metabolism? It is not clear.

-Figure 7: increase the size of the figure text, it is hard to read

-Line 400: TNF-a: do not use bold text

-Line 402: patients, respectively

-Line 407: “gram-negative lipopolysaccharide (LPS)-producing bacteria”

-Line 408: amend “LPS released by these bacteria may mediate inflammation, obesity and insulin resistance”; also, I cannot find LPS information in reference 70, please clarify or remove the reference

-Line 410: “obese prediabetic mice, as well as”

-Lines 413-416: improve punctuation; “vice versa”

-Line 417: ,while…..in both T1DM and T1DM+CD groups.

-Line 419: CD, while decreased in T1DM and T1DM+CD groups

-Line 426: The recent TEDDY (The Environmental Determinants of Diabetes in the Young) study

Lines 428-429: T1DM-associated….CD-associated

-Line 430: remove tTg

-Line 433: islet autoantibodies

-Line 434: in line with

-Line 437: “is distinctly different”: remove distinctly

Lines 439-440: “in presence of high tTg-IgA titers”;; “for the first time”

-Line 458: in disorders characterized by neurodegeneration and neuroinflammation

-Line 461: suggest that ….protective role against neuropathy

-Line 469: retinoic acid modulates?

-Line 471: “mitochondrial dysfunction”

-Lines 469-473: authors should also discuss their results on ethylbenzene degradation

-Line 477: tTG-IgA titers

-Line 478: “Finally, this study suggest a potential role for specific….”

-Line 479: “subclinical neuropathy in pediatric patients with CD and T1D”.

Reviewer 2 Report

The paper is very interesting and the authors open new prespectives of neuropathy diagnosis by corneal nerve fiber damage and the role of gut microbiota in celiac disease and Type 1 Diabetes Mellitus
